# Effect of a Single Bout of Aerobic Exercise on Kynurenine Pathway Metabolites and Inflammatory Markers in Prostate Cancer Patients—A Pilot Randomized Controlled Trial

**DOI:** 10.3390/metabo11010004

**Published:** 2020-12-23

**Authors:** Alexander Schenk, Tobias Esser, André Knoop, Mario Thevis, Jan Herden, Axel Heidenreich, Wilhelm Bloch, Niklas Joisten, Philipp Zimmer

**Affiliations:** 1Institute for Sport and Sport Science, TU-Dortmund, Otto-Hahn-Str. 3, 44227 Dortmund, Germany; alexander.schenk@tu-dortmund.de (A.S.); niklas.joisten@tu-dortmund.de (N.J.); 2Department of Molecular and Cellular Sport Medicine, Institute for Cardiology and Sports Medicine, German Sport University Cologne, Am Sportpark Müngersdorf 6, 50933 Cologne, Germany; tobias95esser@gmail.com (T.E.); w.bloch@dshs-koeln.de (W.B.); 3Institute of Biochemistry, Center for Preventive Doping Research, German Sport University Cologne, Am Sportpark Müngersdorf 6, 50933 Cologne, Germany; a.knoop@biochem.dshs-koeln.de (A.K.); thevis@dshs-koeln.de (M.T.); 4Department of Urology, Uro-Oncology, Robot Assisted and Reconstructive Surgery, University Hospital Cologne, Kerpenerstr. 62, 50937 Cologne, Germany; jan.herden@uk-koeln.de (J.H.); axel.heidenreich@uk-koeln.de (A.H.)

**Keywords:** exercise, physical activity, prostate cancer, kynurenine, tryptophan

## Abstract

The kynurenine (KYN) pathway gains growing research interest concerning the genesis, progression and therapy of solid tumors. Previous studies showed exercise-induced effects on metabolite levels along the KYN pathway. Modulations of the KYN pathway might be involved in the positive impact of exercise on prostate cancer progression and mortality. The objective of this trial was to investigate whether a single-physical exercise alters tryptophan (TRP) metabolism and related inflammatory markers in this population. We conducted a randomized controlled trial with 24 patients suffering from prostate cancer. While the control group remained inactive, the intervention group performed a 30-min aerobic exercise on a bicycle ergometer at 75% of individual VO_2peak_. Before (t0) and directly after the exercise intervention (t1) KYN, TRP, kynurenic acid, quinolinic acid as well as various inflammation markers (IL6, TNF-α, TGF-β) were measured in blood serum. At baseline, the present sample showed robust correlations between TRP, KYN, quinolinic acid and inflammatory markers. Regarding the exercise intervention, interaction effects for TRP, the KYN/TRP ratio and TGF-β were observed. The results show for the first time that acute physical exercise impacts TRP metabolism in prostate cancer patients. Moreover, baseline associations underline the relationship between inflammation and the KYN pathway in prostate cancer.

## 1. Introduction

Prostate cancer, as the second most diagnosed malignancy in men, represents the fifth leading cause of death worldwide [1]. Evidence from prospective randomized controlled trials shows that physical exercise is able to improve patients’ physical capacity and to reduce disease- and treatment-associated side effects, such as lower body strength, fatigue, quality of life and functional performance [2,3]. Moreover, evidence from epidemiological and bench research suggests that physical exercise may also reduce the development and progress of the disease [2,4]. Increased levels of physical activity were proposed to dramatically reduce prostate-cancer-specific and overall mortality [5]. However, the molecular mechanisms underlying the multi-level benefits of exercise in prostate cancer patients remain largely unknown. A broad spectrum of potential underlying mechanisms has been investigated in rodents and cell culture studies, whereas evidence in humans is lacking [6,7]. In order to provide population-specific exercise recommendations, a better understanding of the underlying mechanism of exercise-induced benefits is highly warranted.

A distinction is made between the short-term and long-term effects of physical exercise, especially with regard to the effects of physical training on the immune system [8]. In the long-term, the anti-inflammatory potential is raised by reducing fat mass, increasing anti-inflammatory regulatory T-cells and influencing the pro-/anti-inflammatory cytokine balance in favor of anti-inflammatory cytokines [9,10]. However, short-term effects by a single bout of exercise initially lead to a transient inflammatory reaction [10]. This reaction is reflected by an increase of stress hormones (e.g., epinephrine, norepinephrine and cortisol), cell redistribution and cytokines (e.g., TNF-α, IL-1β and IL6) [11,12,13].

Particularly pro-inflammatory cytokines like tumor necrosis factor (TNF-α), interferon-gamma (INF-γ) and interleukin (IL6) can cause a drastic increase in indoleamine 2,3-dioxygenase (IDO) activity [14,15,16,17]. Together with tryptophan-2,3-dioxygenase (TDO), the two isoenzymes IDO1 and 2 catalyze the initial step of the main metabolic pathway (>95%) of tryptophan (TRP) via the kynurenine (KYN) pathway [18]. In contrast to TDO, which is mainly synthesized and permanently released in the liver, IDO1 is produced in all cell types, mainly as a result of inflammatory stimuli [19,20,21]. Therefore, the ratio of KYN to TRP (KYN/TRP) is often used to indicate changes in the activity of the extrahepatic enzyme IDO [22]. Physiologically, the degradation of TRP via KYN plays a crucial role in the energy supply of the cell by producing nicotinamide adenine dinucleotides (NAD^+^). While dysregulations at different levels of the KYN metabolism have been described in numerous tumor diseases and diseases with a chronic course and are associated with an influence on the course of disease [23,24].

Interestingly, several downstream metabolites along the KYN pathway impact inflammation, immune response and excitatory neurotransmission [25]. Fallarino et al. showed that 3-hydroxyanthranilic and quinolinic acid induce selective apoptosis in vitro in murine thymocytes and Th1 cells, which contributes to T-cell suppression [26]. Furthermore, studies demonstrate that KYN pathway metabolites mediate the differentiation of regulatory T cells, the activity of NK cells and a reduction of cytotoxic immune cells [27,28,29,30].

In addition to other carcinomas, increased IDO activity was observed in patients with prostate cancer [31,32]. Studies in rodents suggest that overexpression of IDO is one of the key mechanisms of local immune suppression and immune eversion in the early stages of tumor progression [33]. IDO1 promotes an inflammatory protumorigenic environment and contributes extensively to neovascularization, which is essential for tumor growth and the formation of metastases [34,35,36]. Based on the increasing knowledge about the influence of IDO-mediated TRP metabolism on the behavior of tumors, research is progressively focusing on the therapeutic manipulation of the key enzymes IDO1 and TDO [24,37].

Exercise has been identified as a modulator of the KYN-pathway. In both healthy and multiple sclerosis patients, it has been shown that an acute (single bout) of aerobic exercise leads to an increased TRP metabolism via the KYN pathway [38,39]. This could provide an explanation for the positive effects of physical activity on the development and progression of miscellaneous diseases. One possible mechanism would be the effect of IL-6 on IDO activity. IL-6 is secreted by the working muscle during exercise and is known to be increased in the circulation after a single exercise bout [40]. Nevertheless, how exactly and on which level the pathway is influenced is not yet clear and should be investigated. 

Recently, two papers have dealt with the effects of exercise on the KYN metabolism in cancer. Zimmer et al. [41] demonstrated that a 12-week resistance exercise program reduced KYN levels in patients with breast cancer undergoing radiotherapy. Similarly, Herrstedt et al. [42] analyzed the effect of a 12-week supervised exercise program in patients with operable gastro-esophageal junction (GEJ) adenocarcinoma undergoing neoadjuvant chemotherapy on plasma concentrations of KYN downstream metabolites and their relation to depressive symptoms.

According to our current state of knowledge, there is no study investigating the influence of a single bout or chronic physical exercise on the KYN pathway in prostate cancer patients. Therefore, the aim of this pilot randomized controlled trial was to investigate whether a single bout of aerobic exercise impacts KYN pathway metabolite levels and related inflammatory markers in prostate cancer patients.

## 2. Results

### 2.1. Participants’ Characteristics

Participants’ characteristics are listed in Table 1. Both groups did not differ in anthropometric, exercise and clinical data at baseline. Regarding clinical data, one patient in the control group revealed very high prostate-specific antigen (PSA) levels and just got a clinical classification of the tumor tissue, because tumor resection was not possible. Missing data were not provided in the medical reports.

### 2.2. Baseline Associations between KYN Pathway Outcomes and Inflammatory Markers

Baseline correlations are presented in Figure 1. KYN was positively correlated with IL-6 (r = 0.544, *p*= 0.009) and negatively correlated with TGF-β (r = −0.485, *p* = *0*.022). KYN was also positively correlated with the ratio of IL-6 to TGF-β (r = 0.630, *p* = 0.002) and the ratio of TNF-α to TGF-β (r = 0.471, *p*= 0.031). The KYN/TRP-ratio, as a marker of KYN pathway activation, was also positively correlated to IL-6 (r = 0.503, *p* = 0.017). Furthermore, quinolinic acid (QA) was positively correlated with IL-6 (r = 0.425, *p*= 0.049). 

### 2.3. Intervention Effect on the KYN Pathway and Inflammatory Markers

Results of the baseline adjusted ANCOVA main effects are presented in Table 2. Regarding the KYN pathway, a significant (time x group) interaction effect for TRP was observed. Post-hoc analyses showed a significant increase of TRP in the control group (CG) (t0: 67.54 ± 12.33 µM; t1: 76.93 ± 13.53 µM; *p* = 0.004) leading to higher values in the CG compared to the intervention group (IG) after exercise (CG: 76.93 ± 13.53 µM; IG: 50.67 ± 7.20; *p* = 0.019). No changes in the IG were detected. A significant (time x group) interaction effect was observed for the KYN/TRP ratio with a significant decline in the CG (t0: 0.022 ± 0.006; t1: 0.019 ± 0.005; *p* < 0.001) with lower values in the CG after exercise (CG: 0.019 ± 0.005; IG: 0.027 ± 0.006; *p* < 0.001). Effects of the intervention on the KYN pathway are presented in Figure 2.

Regarding the inflammatory markers, TGF-β showed a significant time and interaction effect. Post-hoc analyses did not reveal significant changes over time in both groups, but higher values after exercise in the IG (CG: 24.81 ± 5.67 ng/mL; IG: 34.91 ± 10.13 ng/mL; *p* = 0.017). For TNF-α a significant time effect was observed. However, post hoc analyses did not reveal significant changes over time. No significant effects were observed for the TNF-α/TGF-β ratio. Effects of the exercise bout on inflammatory markers are presented in Figure 3.

## 3. Discussion

This is the first study investigating the effect of physical exercise on different KYN pathway metabolite levels and related inflammatory markers in prostate cancer patients. Baseline correlations revealed robust associations between KYN pathway parameters and inflammatory markers in the present sample. Overall, our results show that one hour after a recent meal exercise directly influences the metabolism of TRP in terms of increased tryptophan degradation and a variation of the KYN/TRP ratio compared to the resting control group. 

Baseline association revealed positive correlations of IL6 with KYN as well as with quinolinic acid (QA) (Figure 1). Additionally and in line with this, the ratios IL-6/TGF-β and TNF-α/TGF-β show a positive correlation with KYN, connecting higher inflammatory status with higher KYN concentrations. This is supported by a negative correlation of the anti-inflammatory TGF-β with KYN. These results are in line with the hypothesis that especially the pro-inflammatory cytokines IL-6 and TNF-α mediate an increased activity of IDO [14,15]. To the best of our knowledge, no study has demonstrated this relationship between humoral inflammatory markers and KYN pathway parameters in prostate cancer patients yet.

Regarding the exercise intervention, we detected higher TRP levels at post-intervention in the control group compared to the exercise group (Figure 2). Similar observations have been made in previous studies [38]. However, other trials in rodents and humans showed that an increased catecholamine release induced by exercise leads to increased lipolysis with the release of non-esterified fatty acids (NEFAs) [44,45,46,47,48]. NEFAs displace TRP from its binding with albumin, increasing the amount of free TRP [49]. This apparent discrepancy can possibly be attributed to the previous food-intake as well as the resulting different amounts of circulating insulin. However, both groups received the standard hospital dinner in the clinic about one hour before the first blood draw. The intestinal absorption of TRP as well as an increase of NEFAs by meal may lead to an elevation of free TRP in the blood of the control group. The persistent TRP levels in the intervention group might be explained by increased IDO activity, resulting in greater TRP degradation via the kynurenine pathway. In addition, an increased amino acid demand for protein biosynthesis in the skeletal muscle is conceivable [50]. 

Concerning KYN, we could not see any statistically relevant alterations, although an interaction effect was observed for the KYN/TRP ratio, with lower values in the control group. The decrease in the KYN/TRP ratio in the control group compared to the intervention group can be attributed to the observed elevation of TRP. KYN downstream metabolites and the KYN/TRP ratio are relevant biomarkers in diseases associated with chronic excesses or inflammation, including prostate cancer [51,52]. The TRP breakdown via KYN is considerably influenced by physical exercise. Despite the fact that this pilot study does not provide any substantial mechanisms or consequences regarding the observed effects on metabolite level, it does show the influence of exercise on the KYN pathway in this population. Future exercise intervention studies may build on these preliminary results. 

No other metabolites or ratios of the KYN pathway showed significant changes, neither within nor between the groups. These results are in contrast to previous studies showing that a single bout of aerobic exercise leads to increased production of kynurenic acid (KA) [23]. The small sample size, differences in the investigated sample population or in the applied exercise modalities might be underlying. 

The pro-inflammatory cytokines IL-6 and TNF- α as well as the anti-inflammatory cytokine TGF-β were investigated as potential mediators between the transient inflammatory effect of a single exercise bout and the KYN metabolism [53,54]. While no significant effects were observed for IL-6 and TNF-α, our data showed a significant increase in TGF-β in the intervention group (Figure 3). Similar effects could be demonstrated in young and healthy people after physical exercise [55]. TGF-β is considered to play an ambivalent role in the genesis and progression of prostate cancer. Due to its anti-proliferative, pro-apoptotic and anti-angiogenetic properties, TGF-β initially acts as a tumor suppressor [56]. Our participants suffer from later tumor stages. In this case, it acts as a tumor promoter due to its immunosuppressive effect, especially in the tumor microenvironment by directly inhibiting immune responses and stimulating angiogenesis [57,58]. However, TGF-β values were measured in the circulation and not in the tumor microenvironment. Therefore, it is unclear whether the serum TGF-β values represent the microenvironment of the tumor tissue.

Potential limitations of our study result from the experimental design and the performed measurements. The small sample size surely limits the findings. Furthermore, baseline-testing also should include measurements of inflammatory markers that could be included as a stratification factor for randomization. However, we performed this pilot study in order to capture potential effects on the biomarkers investigated and gain knowledge for future powered randomized controlled trials with chronic exercise interventions. Furthermore, testing of aerobic capacity was performed with 20 W steps in all participants as recommended for cancer patients [59]. However, three patients exceeded exercise testing time and exercise testing termination could be due to fatigue and does not reflect peak exercise capacity. Nevertheless, this reflects a minority of patients and the exercise testing protocol was suitable for the majority of patients.

Despite the fact that we have used the gold standard method (high-performance liquid chromatography (HPLC) and mass spectrometer (MS)) to determine all central metabolites in blood serum, future studies should include further mediators (INF-γ, Cortisol) and enzyme expressions (IDO, TDO, KMO) in tissues of interest, for example, immune cells or skeletal muscle. Badaway et al. [60] recently discussed different determinants influencing the KYN/TRP ratio that consequently limit the diagnostic value of this parameter with regard to the activity of IDO. Although all participants received a meal at the same time of the day, standardized prepared meals would reduce the influence of different nutritional uptake.

In conclusion, this is the inaugural study investigating the effect of a single bout of physical exercise in prostate cancer patients on various KYN parameters and selected inflammation markers. We demonstrate that inflammation markers correlate significantly with KYN pathway metabolites. Moreover, the intervention showed effects on both inflammation cytokines and TRP-concentration. The decrease in free TRP is likely mediated by activation of IDO. Future randomized controlled trials should investigate chronic effects on inflammation and KYN metabolism in prostate cancer patients using larger populations and exercise programs lasting several weeks.

## 4. Materials and Methods 

This pilot study was approved by the ethics committee of the University Hospital Cologne and accords to the Declaration of Helsinki. The study was prospectively registered at the German Clinical Trials Register (DRKS00010442). All patients were briefed and provided written informed consent prior to baseline testing.

In this randomized controlled trial, 24 prostate cancer patients conducted either a single bout of aerobic exercise or remained sedentary for the same period of time as a passive control. Inclusion and exclusion criteria are presented in Table 3. All participants performed a graded exercise test (GXT) on a bicycle ergometer for assessment of maximal oxygen consumption (VO_2peak_) during baseline testing. Subsequently, participants were randomized with concealed allocation into the intervention or control group using the minimization according to Pocock and Simon using the maximal oxygen consumption as a stratification factor. One week after the baseline testing, participants performed the intervention and control session, respectively. Blood samples were drawn from all participants immediately before and immediately after the intervention/control session.

### 4.1. Baseline Testing

Patients were recruited prior to their tumor resection and stationary stay in the hospital. One week prior to hospitalization, patients underwent baseline testing with the recording of anthropometric data and assessment of the maximal oxygen consumption (VO_2peak_) using a GXT. The GXT was performed on a stationary bicycle ergometer (Ergoline) and VO_2peak_ was measured using a METALYZER^®^ 3B (CORTEX Biophysik GmbH, Leipzig, Germany). Using a step protocol, patients started at 20 W power output and the workload was subsequently increased by 20 W every 2 min until exhaustion [59]. Patients were encouraged to hold a cadence between 70–80 rpm. Heart rate was measured during the GXT with a heart frequency sensor (Polar Electro GmbH Deutschland, Büttelborn, Germany) and the level of subjectively perceived exhaustion was assessed using the Borg scale [61]. Exhaustion was determined by a decrease of cadence below 60 rpm, respiratory quotient (VCO_2_/VO_2_) of 1.1 or subjectively perceived exhaustion (Borg value > 18).

### 4.2. Intervention

The intervention was conducted on the day of hospitalization. All patients entered the hospital in the morning and passed through medical assessments without any amount of physical arousal during the day. Dinner was served at 06:00 PM by the hospital while the intervention was conducted at 07:00 PM. The IG performed a single bout of endurance exercise on a bicycle ergometer (Ergoline) at moderate to vigorous intensity. The exercise bout started with an individual warm-up for 5 min. Subsequently, the patients cycled for 20 Min at a resistance of the ergometer corresponding to the resistance at 75% of the individual VO_2peak_ during the GXT baseline test. Finally, an individual cool down for 5 min was performed. The CG stayed sedentary during the same period of time.

### 4.3. Blood Sampling

Blood samples were drawn before and directly after cessation of the 30 min exercise bout (IG) or after remaining sedentary for 30 min (CG). After clotting, serum was collected by centrifugation at 1100× *g* for 10 min. Thereafter, serum was aliquoted and stored at −80°C until sample measurement.

### 4.4. Assessment of KYN Pathway Metabolites

The KYN pathway metabolites TRP, KYN, QA and KA were measured via HPLC coupled to an MS. According to prior published sample preparation [23], serum was stored in 50 µL aliquots at −80°C until analysis. For the analysis, internal standards (ISTD) were mixed with the serum samples. ISTD consists of deuterium labeled TRP, KYN, QA and KA. To remove proteins from the serum samples, ice-cold methanol was added and mixed vigorously. Crashed proteins were removed by centrifugation for 5 min at 17.000× *g*. The supernatant was transferred into an HPLC glass vial to be injected into the HPLC-MS system. A calibration curve for each analyte was prepared in artificial serum that consists of 1% human serum albumin (Biotest Pharma GmbH, Dreieich, Germany). 

The analysis was performed on a Waters ACQUITY UPLC^®^ system equipped with an ACQUITY UPLC^®^ HSS T3 analytical column coupled to a Xevo^®^ TQ-XS triple quadrupole mass spectrometer (Waters, Eschborn, Germany). For the chromatographic separation, a flowrate of 300 µL/min was set. As eluent A a 5 mM ammonium acetate solution with pH 9 was used. As eluent B acetonitrile acidified with 2% formic acid was used. The injection volume was 2 µl. Positive ionization was performed by an UniSpay™ ion source with an impact voltage of 3.9 kV at 500 °C. Multiple reaction-monitoring experiments were conducted for ion transitions resulting from collision-induced dissociations in the presence of Argon. The concentrations were calculated by the peak area ratios of the quantifying ion transition and the corresponding signal for the ISTD using a calibration curve.

### 4.5. Assessment of Inflammatory Markers

The inflammatory cytokines IL-6, TNF-α and TGF-β were measured using commercial ELISA-kits (R&D systems, Minneapolis, USA) according to the manufacturer’s protocol.

### 4.6. Statistics

To evaluate baseline associations between TRP metabolites and (anti-)inflammatory markers in this sample of prostate cancer patients, Spearman’s correlation coefficients were calculated. Subsequently, we checked our sample for potential baseline differences regarding anthropometric and performance-related characteristics between the groups using an independent *t*-test.

In order to assess differences within and between groups in KYN pathway outcomes pre- and post-intervention, baseline-adjusted analyses of variances (ANCOVA) were conducted. In case of significant ANCOVA main effects, Bonferroni corrected post-hoc comparisons were conducted to identify statistically significant main effects within the group effects. The level of significance was set at *p* ≤ 0.05. All statistical analyses were conducted using IBM SPSS Statistics (Version 27).

## Figures and Tables

**Figure 1 metabolites-11-00004-f001:**
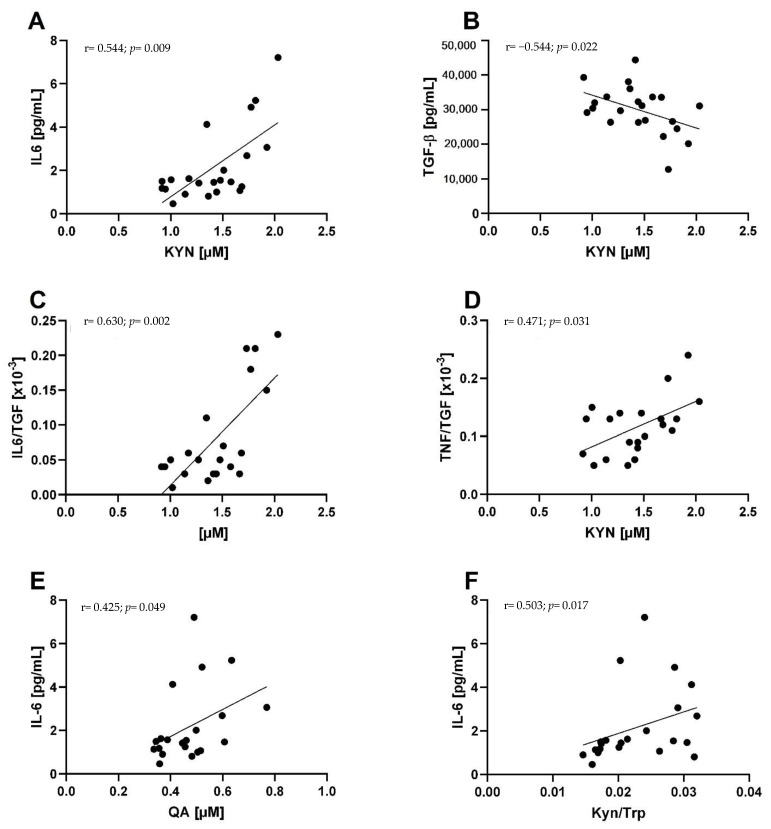
Baseline associations between the KYN pathway and inflammatory markers. Correlations were calculated according to Spearman’s coefficients. Graphs show the correlations of KYN with IL-6 (**A**), TGF-β (**B**), as well as the ratios of IL-6/TGF-β (**C**) and TNF-α/TGF-β (**D**). Furthermore, the correlations of IL-6 with QA (**E**) and the KYN/TRP-ratio (**F**) is shown. Correlation coefficient and *p*-values are provided in each diagram.

**Figure 2 metabolites-11-00004-f002:**
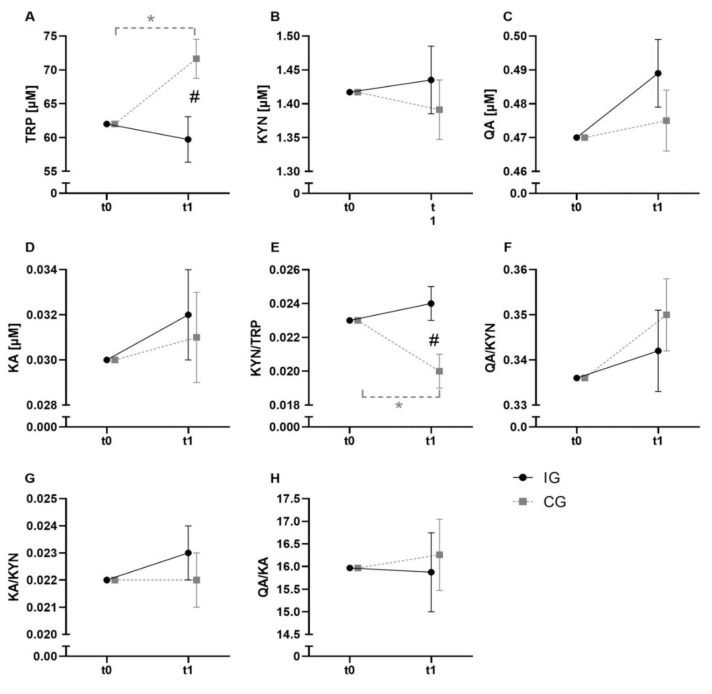
KYN pathway results of the baseline-adjusted ANCOVA. Diagrams show the baseline adjusted serum concentrations of the KYN-pathway metabolites TRP (**A**), KYN (**B**), QA (**C**) and KA (**D**). Furthermore, the ratios of KYN/TRP (**E**), QA/KYN (**F**), KA/KYN (**G**) and QA/KA (**H**) are shown.IG is presented in black color and solid lines, whereas CG is in grey and broken lines. Significant post-hoc results for time effects are indicated by an asterisk (*) and for interaction effect by “#”. The significance level was set to α ≤ 0.05.

**Figure 3 metabolites-11-00004-f003:**
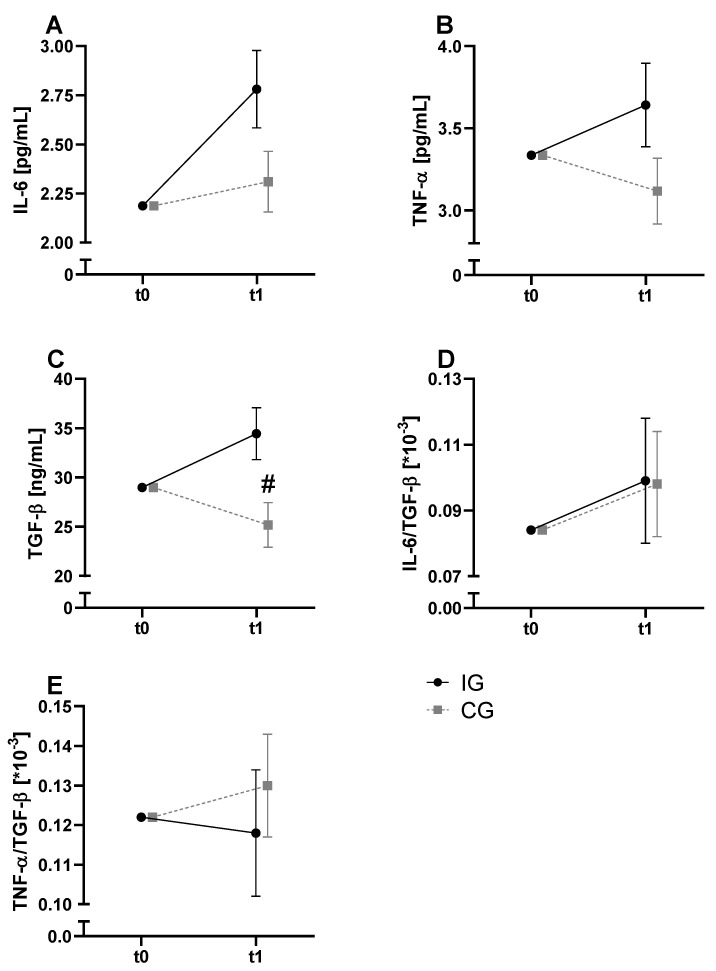
Results of the baseline-adjusted ANCOVA for inflammatory markers. Baseline-adjusted serum levels of IL-6 (**A**), TNF-α (**B**) and TGF-β (**C**) before and after exercise are presented. Furthermore, the ratios of IL-6/TGF-β (**D**) and TNF-α/TGF-β (**E**) are shown. IG is presented in black color and solid lines, whereas CG is in grey and broken lines. Significant post-hoc results for interaction effect by “#”.

**Table 1 metabolites-11-00004-t001:** Participants’ characteristics. Data are presented as mean ± SD. A *t*-test was used for group comparison of metric data and Pearson-χ²-test for normative data (Gleason score, International Society of Urological Pathology (ISUP) classification, risk classification according to D’Amico [43] and tumor state). RQ and Borg values are the maximal achieved values during the baseline graded exercise test (GXT). HR = heart rate, RQ = respiratory quotient.

	Overall (*N* = 24)	Intervention group (*n* = 11)	Control group (*n* = 13)	*p*-Value
Age [years]	64.9 ± 8.4	64.6 ± 7.9	65.6 ± 7.8	0.927
BMI [kg/m²]	26.3± 3.7	24.7 ± 2.9	27.6 ± 3.9	0.057
HR_max_ [bpm]	145 ± 29	146 ± 33	142 ± 24	0.734
Max Power output [W]	148 ± 40	148 ± 40	151 ± 44	0.994
VO_2peak_ [ml/min/kg]	25.5 ± 6.3	27.2 ± 6.3	25.5 ± 6.9	0.355
RQ_max_	1.10 ± 0.06	1.12 ± 0.05	1.10 ± 0.6	0.337
Borg_max_	17 ± 2	17 ± 2	16 ± 2	0.515
Gleason score	7 (*n* = 2)	7 (*n* = 0)	7 (*n* = 2)	0.474
	7a (*n* = 8)	7a (*n* = 4)	7a (*n* = 4)	
	7b (*n* = 6)	7b (*n* = 2)	7b (*n* = 4)	
	8 (*n* = 4)	8 (*n* = 2)	8 (*n* = 2)	
	9 (*n* = 4)	9 (*n* = 3)	9 (*n* = 1)	
ISUP classification	2 (*n* = 8)	2 (*n* = 4)	2 (*n* = 4)	0.644
	3 (*n* = 6)	3 (*n* = 2)	3 (*n* = 4)	
	4 (*n* = 4)	4 (*n* = 2)	4 (*n* = 2)	
	5 (*n* = 4)	5 (*n* = 3)	5 (*n* = 1)	
	Missing (*n* = 2)	Missing (*n* = 0)	Missing (*n* = 2)	
PSA level [ng/mL]	18.70 ± 29.04	14.10 ± 9.47	22.59 ± 38.80	0.488
Risk classification	High risk (*n* = 21)	High risk (*n* = 11)	High risk (*n* = 10)	0.089
	Intermediate risk (*n* = 3)	Intermediate risk (*n* = 0)	Intermediate risk (*n* = 3)	
Tumor state	cT2c (*n* = 1)	cT2c (*n* = 0)	cT2c (*n* = 1)	0.549
	pT2a (*n* = 2)	pT2a (*n* = 0)	pT2a (*n* = 2)	
	pT2b (*n* = 1)	pT2b (*n* = 0)	pT2b (*n* = 1)	
	pT2c (*n* = 11)	pT2c (*n* = 6)	pT2c (*n* = 5)	
	pT3 (*n* = 1)	pT3 (*n* = 1)	pT3 (*n* = 0)	
	pT3a (*n* = 4)	pT3a (*n* = 2)	pT3a (*n* = 2)	
	pT3b (*n* = 4)	pT3b (*n* = 2)	pT3b (*n* = 2)	

**Table 2 metabolites-11-00004-t002:** Intervention effect on the KYN pathway and inflammatory markers. Data are presented as mean ± SD. The main effects of the baseline adjusted ANCOVA are time and interaction effects.

Parameter	Group	Point in Time		ANCOVA Time			ANCOVA Interaction (Time × Group)		
		T0	T1	*p*	df	F	*p*	df	F
TRP [µM]	IG	51.2 ± 8.2	50.3 ± 8.0	0.073	1	3.626	0.019	1	6.608
	KG	68.1 ± 12.9	78.6 ± 13.0						
KYN [µM]	IG	1.3 ± 0.2	1.3 ± 0.1	0.594	1	0.295	0.523	1	0.425
	KG	1.5 ± 0.4	1.5 ± 0.4						
QA [µM]	IG	0.4 ± 0.05	0.5 ± 0.04	0.406	1	0.725	0.293	1	1.173
	KG	0.5 ± 0.13	0.5 ± 0.13						
KA [µM]	IG	0.03 ± 0.006	0.03 ± 0.006	0.724	1	0.129	0.614	1	0.265
	KG	0.03 ± 0.011	0.03 ± 0.011						
KNY/TRP ratio	IG	0.03 ± 0.006	0.03 ± 0.006	0.189	1	1.863	<0.001	1	22.790
	KG	0.02 ± 0.006	0.02 ± 0.005						
QA/KYN ratio	IG	0.34 ± 0.03	0.34 ± 0.04	0.426	1	0.662	0.484	1	0.511
	KG	0.34 ± 0.04	0.35 ± 0.06						
KA/KYN ratio	IG	0.02 ± 0.004	0.02 ± 0.005	0.842	1	0.041	0.867	1	0.029
	KG	0.02 ± 0.004	0.02 ± 0.005						
QA/KA ratio	IG	16.8 ± 2.6	15.6 ± 2.3	0.509	1	0.456	0.747	1	0.107
	KG	16.6 ± 3.6	16.9 ± 4.5						
IL-6 [pg/mL]	IG	2.0 ± 1.3	2.8 ± 1.5	0.093	1	3.148	0.078	1	3.48
	KG	2.5 ± 2.2	2.6 ± 2.1						
TGF-β [ng/mL]	IG	30.5 ± 5.0	34.9 ± 10.1	0.028	1	5.735	0.017	1	6.906
	KG	27.8 ± 7.0	24.8 ± 34.9						
TNF-α [pg/mL]	IG	3.4 ± 1.1	3.7 ± 0.7	0.012	1	7.88	0.124	1	2.608
	KG	3.4 ± 1.2	3.3 ± 1.1						
TNF-α/TGF-β ratio [x10^−5^]	IG	10.6 ± 4.0	11.2 ± 3.2	0.004	1	9.914	0.585	1	0.457
	KG	13.1 ± 6.0	14.1 ± 5.4						
IL-6/TGF-β ratio [x10^−5^]	IG	6.0 ± 3.3	8.0 ± 3.7	0.627	1	0.293	0.967	1	0.003
	KG	10.2 ± 8.7	11.4 ± 10.9						

**Table 3 metabolites-11-00004-t003:** In- and exclusion criteria.

Inclusion Criteria	Exclusion Criteria
Diagnosed high- or intermediate-risk prostate cancer (PSA > 10 ng/mL or Gleeson score ≥ 7 or cT2b) Age above 18 years Planned tumor resection	Chemotherapy or radiation prior to study participation A previous cancer disease Diseases of the lung, cardiovascular system or orthopedic issues precluding participation

## Data Availability

The data presented in this study are available on request from the corresponding author.

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
