# Peer review of "Effect of a Single Bout of Aerobic Exercise on Kynurenine Pathway Metabolites and Inflammatory Markers in Prostate Cancer Patients—A Pilot Randomized Controlled Trial"

_metabolites, 2020, doi:10.3390/metabo11010004_

Round 1

Reviewer 1 Report

Schenk et al. present a paper examining the acute effect of aerobic exercise on tryptophan metabolites in prostate cancer patients. In particular, the authors hoped to explore alterations with exercise in the KYN pathway, which has a role in tumor progression. Secondary study outcomes included changes in inflammatory markers. To accomplish these study objectives, prostate cancer patients were recruited and randomized to either an acute bout of exercise or a control sedentary situation. Study findings included positive correlations with tryptophan and inflammatory markers at baseline as well as changes with tryptophan with exercise training. The authors concluded that this is the first study in humans to show that an acute bout of exercise impacts tryptophan metabolism in prostate cancer patients. The study is strong due to the randomization study design. However, it has some minor issues to address before consideration for full acceptance. 

Major Comments

  • Since this study has an emphasis on inflammation, it is not clear to me why these data are not included in Table 1. Were there differences in inflammatory markers between the two groups at baseline? It seems to me like this was not measured at the baseline visit (no blood draw), but should at least be discussed as a limitation in the study.
  • Was the data tested for normality? What is the statistical reasoning for using non-parametric Spearman correlations? 

Minor Comments

  • Introduction line 36, the "s" is not needed after the word "shows".
  • Introduction line 44, "has" should be the word "have". 
  • Introduction line 58, TRP is not defined yet in the article.
  • There are also many inconsistencies in the introduction between the use of word abbreviations. For example, line 82, kynurenine is spelled out whereas it is abbreviated in like 83. Please be consistent.
  • Results, Section 2.2. Please consider moving the sentence in line 108, "Baseline correlations are presented in Figure 1" to the beginning of that paragraph. 
  • Figure 2 legend: IG and CG are not defined.

Reviewer 2 Report

This manuscript addresses an interesting area of research. However, several issues were raised while reviewing and please see below for the specific comments.

Title needs to be changed to reflect this is a single bout of aerobic exercise. Also, using ‘a single bout of aerobic exercise’ or similar terminology throughout the manuscript would be recommended as ‘acute exercise bout’ is not clearly showing what the authors conducted in this manuscript.

Abstract

  1. Please present the duration of a bout of aerobic exercise (in minute?, rpm or power output)
  2. Between group comparisons need to be reported in the abstract. Currently, authors only state interaction effects for TRP, the KYN/TRP ratio and TGF-b were observed. This is very unclear what the response to a single bout of exercise is. Compared to control group, what was significantly increased or decreased? These need to be reported.
  3. Justification about prostate cancer needs to be presented. Why is this mechanism is important in prostate cancer, rather than any other populations?

Introduction

Please be consistent using either prostate cancer or prostate carcinoma.

Line 83: several weeks sound vague, please specify the exact duration.

Obviously, there are different time points; before, during, after prostate cancer treatment. Where was the target prostate cancer patient in?

Table 1. total sample here shows n=24 and abstract shows 28. Unclear what happened. ISUP, PSA need to be defined in the title of table 1. As for the BMI, both groups are close to 25.0 but overall 26.9? Also p-value shows a statistical trend while both BMI is almost identical. Please double check if statistical analysis was done properly. If this was done properly, BMI may need to be included in ANCOVA model and see if it influences the data.

2.2. Was there any correlation with VO2peak data and blood markers?

2.3. “Significant interaction” was group x time interaction? If so, please clarify. Also, why was there a significant increase of TRP in control group? Shouldn’t the control group stay the same as they didn’t exercise? If this is true, the blood assay would not be reliable as it is easily changed when patients did ‘nothing’. This is very much questionable for all markers presented here.

Table 2. This is very unorganized. Also df and F value may not need to be reported. Rather, raw value (pre and post) and p-value should be reported in each group. Please revise.

Material and methods

  1. What was the VO2max exercise protocol? Ramp protocol? or any other?
  2. Unclear what high and intermediate risk prostate cancer. How was this defined?
  3. All prostate cancer included in the study was planned to receive tumor resection? How long after the study?
  4. Exercise protocol seems 20W ramp protocol for everyone, which may be problematic. Each patient has a different aerobic capacity and if 20W is applied to everyone, it can induce early termination of exercise testing for someone. Also, unclear how 60rpm is cut-off point for the definition of exhaustion.
  5. Blood was centrifuged at 1100 xg for 5min. Please show reference on this. Also was this fasting blood or not? Seems like it is not as the intervention as conducted at 7PM.
  6. How the power output corresponding to 75% was ensured? Please clarify.

Statistics.

Why was spearman was the only correlation method?

What were the adjusted variables in the ANCOVA model?

How was sample size of 28 or 24 determined?

Reviewer 3 Report

Introduction

line 58: TRP has not been previously defined in the body of the manuscript, only the abstract

lines 63-68: This paragraph needs a conclusion statement to indicate the final point and to transition to the next paragraph.

line 78: instead of general and plural term "diseased populations," may be better to specifically indicate people with multiple sclerosis since that is the only one and is not a type of cancer, which may be expected as a comparison since the topic of the current paper is a cancer

comment pertaining to lines 58-62, lines 74-75, and lines 77-79: it is still unclear what the role of TRP in prostate cancer is, and why you would want to use exercise to increase its metabolism. Make these connections explicit.

lines 83-87: Even though the numerical citations were both referenced at the end of the first sentence in this paragraph, please include the citations again individually to go along with each study

lines 89-91: Unclear exactly what the focus is: title indicates tryptophan, line 89 indicates KYN; I understand they are related, but the lack of clarity of purpose ties in with the need for more explicit connection of the role of TRP

Results

Table 1: What is the risk classification referring to? Later on in the inclusion/exclusion criteria I see, but by Table 1 this is not clear.

Discussion

line 143: Do you need to specifically bring up that the tryptophan degradation occurred after a recent meal?

Methods

line 234: Did you mean to write decadence?

line 239: Why was the intervention conducted one hour after dinner was served? Was the dinner a standardized meal? Was the dinner high in tryptophan? Do you know what the actual time between the participants finishing their meal and starting exercise was? Could different amounts of circulating insulin have impacted the study's results? Especially after reading lines 159-161 in the discussion, a justification of a recent meal prior to testing is warranted.

line 252: states "according to prior published sampled preparation," but does not cite that prior publication

Round 2

Reviewer 2 Report

Thank you for the edits and response. While there was an attempt to address the limitation of exercise protocol (20W), it still does not make sense, especially due to the duration reported now.

If the mean duration was 15minutes (9-25minutes), it does not accurately reflect true max capacity. It appears the range is up to 25 minutes which is terminated by fatigue rather than regular physiological response. This means the workload was mistakenly set up.

Author Response

Dear Reviewer,

we appreciate your comments. The exercise testing protocol was chosen in regard to the recommendations for cancer patients by Sharhag-Rosenberger et al. (2014). The optimal time interval of 9-16 Minutes for a step-protocol was exceeded by a minority of three patients. In regard of this three patients the termination of exercise may was due to fatigue. We added this to the limitations.
